# A Case of Cushing's Syndrome from Well-Differentiated Neuroendocrine Tumors of the Small Bowel and Its Mesentery

**Kirsten Rose Carlaw \***, **Ahmer Hameed and Anthony Shakeshaft**

Colorectal Surgery Department, Nepean Hospital, Kingswood, NSW 2747, Australia
\* Correspondence: kirstenrosecarlaw@gmail.com

**Abstract:** Adrenocorticotropic (ACTH)-producing neuroendocrine tumours (NETs) are rarely found in the small bowel, and primary mesenteric NETs have only been reported in a few cases globally. We report the case of a 68-year-old female with ectopic Cushing's syndrome due to excessive ACTH secretion from small bowel primary lesions and mesenteric metastasis. Initially, only the mesenteric mass was detected on imaging and endoscopy/colonoscopy, and it was only with surgical exploration that the small bowel lesions were found. This highlights the importance of high clinical suspicion and robust investigation when locating NETs. Surgical resection of the affected small bowel and mesentery was the definitive treatment for this patient. Initial hydrocortisone replacement therapy was needed, and subsequent biochemical tests and clinical reviews demonstrated no recurrence.

**Keywords:** ectopic ACTH; neuroendocrine tumour; NET; mesenteric NET; small bowel NET





## 1. Introduction

Neuroendocrine tumours (NETs) are derived from neuroendocrine cells and commonly involve the lungs, i.e.,bronchial carcinoma, and the gastrointestinal system, particularly the pancreas but rarely the ileum, mesentery, and liver [1]. Mesenteric NETs are extremely rare and often secondary to another primary tumour [2]. NETs can cause unregulated production of adrenocorticotropic hormone (ACTH) and subsequent hypercortisolism resulting in ectopic Cushing's syndrome [1]. Clinical manifestations of Cushing's syndrome vary in severity and include hypokalemia, cardiovascular complications, acute respiratory distress, infectious complications, and bone fractures [1]. The diagnosis of NETs can be challenging and consists of biochemical and imaging investigations and more invasive procedures. Imaging techniques include ultrasound, CT chest/abdomen/pelvis, and functional imaging such as octreotide scans, PET/CT DOTATATE scans, or FDG PET/CT [2].

Our aim is to discuss a diagnostically challenging case of small bowel and mesenteric ACTH-secreting NETs that was initially on imaging and endoscopy/colonoscopy thought to be a primary mesenteric lesion but intraoperatively and on histopathology was found to be likely secondary to a small bowel NET.

## 2. Case Presentation

A 68-year-old female was referred by her general practitioner to hospital for critical hypokalemia of 1.7, metabolic alkalosis, and hypertension (180/90 mmHg). Her symptoms included a week of insomnia, intermittent cardiac palpitations at night, unstable balance, and a metallic taste in her mouth. Upon examination, she had cushingoid features of a buffalo hump and central adiposity and no lymphadenopathy. She had a past medical history of recently diagnosed chronic lymphocytic leukemia (CLL), ischemic heart disease, and lower limb deep vein thrombosis (DVT) for which she was taking daily rivaroxaban. She had no significant family history and was relatively active and independent in the activities of daily living. Written informed consent was obtained from the patient to publish this paper.

Investigations and management were guided by the endocrinology team. Laboratory testing showed a normal aldosterone:renin ratio, a high ACTH level of 36.6 pmol/L (normal range: 1.0–10.8), high morning cortisol levels of 1869 nmol/L, increased 24 h urinary free cortisol 4349 nmol/24 h (normal range < 166), and a positive low/high dose dexamethasone suppression test were indicative of ectopic ACTH. She was commenced on oral metyrapone 750 mg TDS which successfully reduced the cortisol level to 319 nmol/L. Tumour markers including AFP, CEA, CA 125, and CA 19.9 were negative. The HbA1c, blood sugar levels, and thyroid function tests were all within normal limits. Her hypokalemia was managed with intravenous and oral potassium replacement and spironolactone. Her hypertension improved with low-dose ramipril.

MRI of the brain revealed no pituitary mass or intracranial pathology. The CT chest/abdomen/pelvis was unremarkable and demonstrated no lymphadenopathy or splenomegaly. The thyroid US was unremarkable. A PET Gallium DOTATATE scan demonstrated a DOTATATE avid mass measuring 4.7 × 2.8 cm in the mesentery suspicious for DOTATATE avid malignancy. There were no other suspicious lesions. This is shown in Figures 1 and 2.

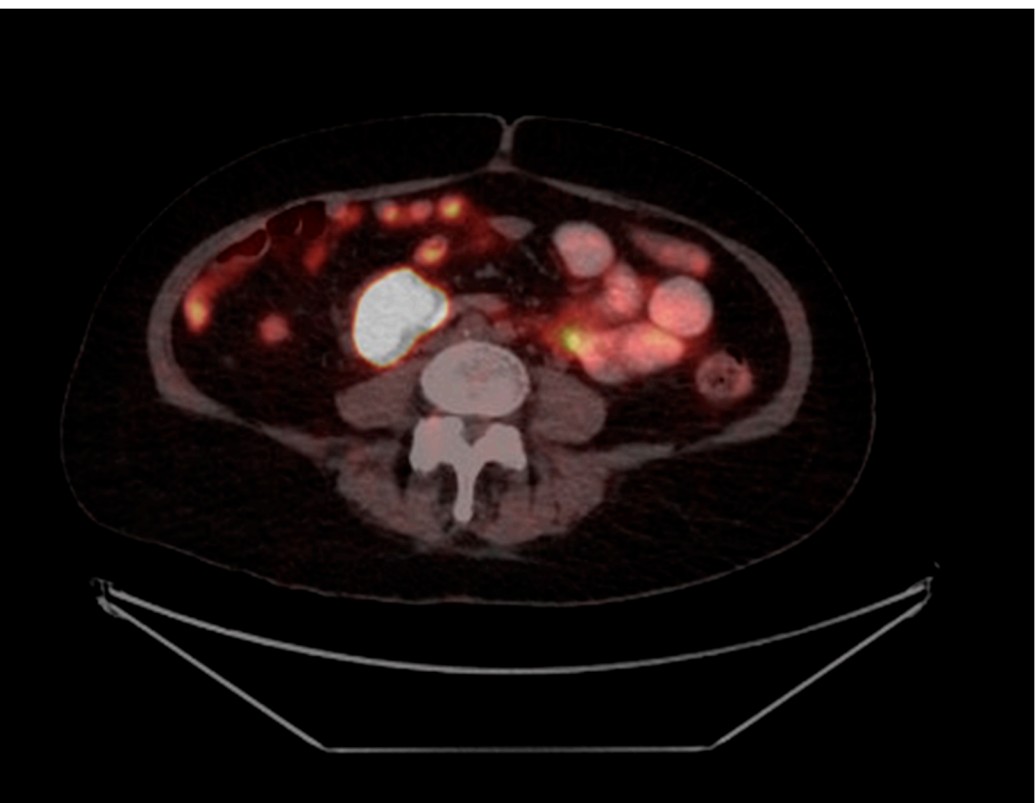

**Figure 1.** CT axial slice of the abdomen as part of the PET Gallium DOTATATE scan showing DOTATATE uptake of the mesenteric mass 47 × 28 mm.

The case was referred to the colorectal department for consideration of surgical resection of the mesenteric mass. A multidisciplinary team discussion with the oncologist, endocrinologist, and surgical teams determined that an endoscopy and a colonoscopy should be performed prior to resection of the mass to further investigate for a possible primary lesion as the consensus was towards the higher probability of a metastatic mesenteric mass even though no primary was found on imaging. The endoscopy demonstrated a normal oesophagus, a regular Z-line, a normal duodenum, and a single gastric polyp that was biopsied demonstrating only moderate chronic inflammation and reactive changes. The colonoscopy was unremarkable. A laparoscopic small bowel resection was performed, and two small bowel nodules were found that were 30mm apart and a mesenteric nodule

was located in the distal third of the small bowel mesentery. This is shown in Figure 3. Nodule 1 was 5 × 5 × 1 mm and nodule 2 was 9 × 8 × 2 mm. Post-operative care was further guided by the endocrinology team including intravenous hydrocortisone that was transitioned to an oral regimen upon discharge.

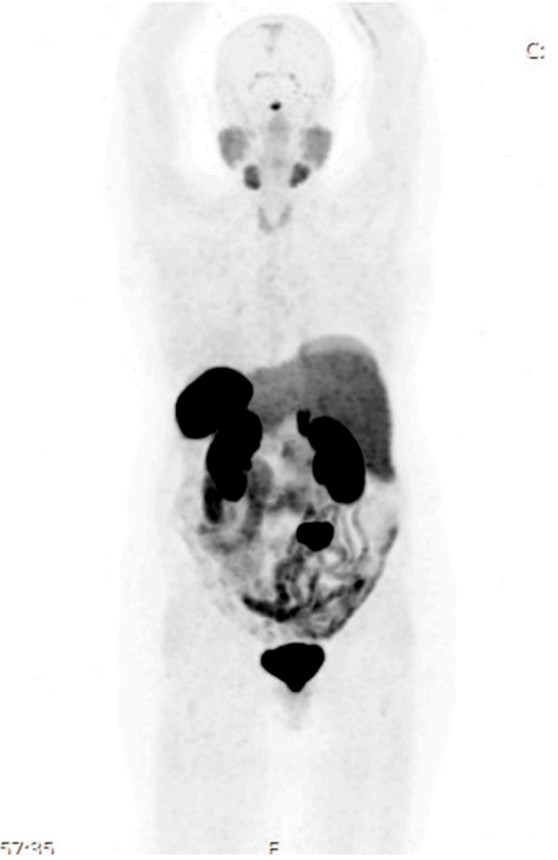

**Figure 2.** Coronal plane view of the PET Gallium DOTATATE Scan showing DOTATATE uptake of the mesenteric mass 47 × 28 mm.

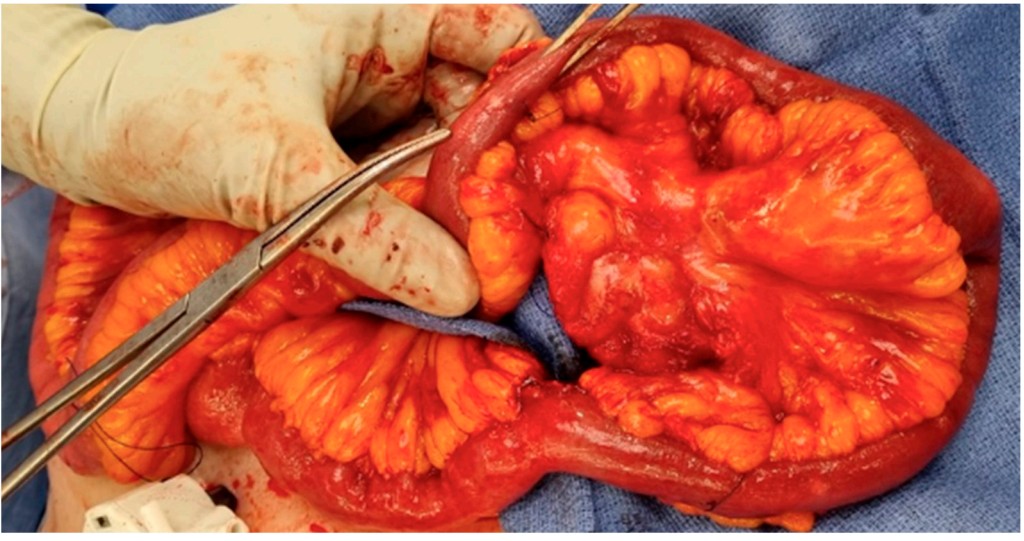

**Figure 3.** Clinical photograph of a section of the small bowel with the mesentery resected, and Roberts forceps indicating the small bowel nodule.

The histopathology revealed that both the small bowel nodules and the mesenteric mass were G1 well-differentiated neuroendocrine tumours of pT4 and pN2. The nodules invaded the visceral peritoneum and were associated with two regional lymph nodes. The surgical margins were clear. The immunohistochemistry showed that the tumours were strongly and diffusely positive for chromogranin, synaptophysin, and CDX2. T1F1 and GATA3 were both negative. Nodule 1 was ACTH-, glucagon-, and gastrin-negative, but somatostatin-isolated-positive cells equalled 1%. Nodule 2 was gastrin- and glucagon-negative; ACTH was scattered positively by only 2%, and somatostatin-isolated-positive cells equalled 1%. The mesenteric mass was ACTH-positive in 50–60% of cells, glucagon-negative, somatostatin-isolated-positive cells equalled <1%, and it was gastrin-negative.

Upon follow-up three months later, the patient was well, had ceased all hydrocortisone replacement, and their 1mg dexamethasone suppression test showed normal ACTH < 1.1 and cortisol < 28. There was no evidence of residual ectopic ACTH production.

## 3. Discussion

Ectopic Cushing's syndrome from ACTH-producing NETs commonly originates from intrathoracic small cell carcinomas, medullary thyroid carcinoma, phaeochromocytoma, and pancreas and thymus carcinomas [3]. More uncommon locations include the appendix, duodenum, ileum, uterine cervix, and ovaries [4]. Evidently, these endocrine tumours can develop from many different tissue origins which highlights the difficulty in classifying and diagnosing them. Our case explores ectopic Cushing's syndrome from rare ACTH-producing NETs of the ileum with associated mesenteric metastases. Our report emphasises the challenges of diagnosing NETs and the need for medical and operative management in the case of NETs affecting the intestine and/or mesentery.

Diagnosis of ACTH-producing NETs includes blood tests particularly high serum ACTH levels with the dexamethasone suppression test, high serum cortisol, and urinary-free cortisol [1]. Clinical signs that indicate Cushing's syndrome as a consequence of hypercortisolism include purple striae, osteoporosis, profound hypokalemia, facial rounding, and severe hypertension with oedema and the absence of weight gain or weight loss [1]. Imaging investigations including CT chest/abdomen/ pelvis, octreotide scans and, if indicated, MRI scans of the pituitary gland are commonly used to identify the location of the NET [5]. 18 FDG-PET scans have been shown to be an effective second-line diagnostic procedure when other imaging techniques fail to localise the NET [6]. In our case, the use of CT chest/abdomen/pelvis and a Ga-68 DOTATATE scan helped to detect the mesenteric mass. Ga-68 DOTATATE as a PET tracer has been shown to have a higher sensitivity and specificity than $^{111}$In-DTPA-Octreotide in detecting NETs [7].

Given the rarity of a mesenteric mass being the primary NET, other diagnostic modalities such as a colonoscopy, small bowel series, and scintigraphy can be used [8]. In our study, the colonoscopy was unremarkable, and it was only by surgical exploration that the small bowel lesions were found. Studies have shown that radio-guided surgery has been diagnostically and therapeutically successful for NETs whereby radiolabeled pentetreotide was injected pre-operatively and a gamma counter probe was used intraoperatively to detect positive sites with uptake [9].

NETs have specific immunohistochemistry findings whereby somatostatin, synaptophysin, chromogranin A, and neuron-specific enolase are usually positive [10]. In our case, the tumours were strongly and diffusely positive for chromogranin, synaptophysin, and CDX2. Histopathology is important as tumour aggressiveness and the intensity of cortisol secretion are not always correlated. Furthermore, well-differentiated ectopic ACTH-secreting NETs can demonstrate slow progression despite being evidently metastatic at the time of presentation. Thus, histopathology of a biopsied lesion is important to determine the differentiation status and proliferation index in this situation.

The curative management for ectopic ACTH-secreting NETs is surgical resection of the tumour [3]. Medical treatment of Cushingoid syndrome prior to the excision of the NET is very important. In our case, metyrapone was used to reduce cortisol levels. Metyrapone

is a specific inhibitor of 11-hydroxylase and is very effective in the short term causing a significant reduction in urinary cortisol within 24 to 72 h. However, it needs to be administered in three to four daily fractions given its short half-life [11]. Other agents such as mitotane, ketoconazole, and aminoglutethimide can also be used. Ketoconazole in particular is a steroid synthesis inhibitor that acts on several cytochrome P450 steroidogenic enzymes and has been shown to reduce serum cortisol levels within a few days [12]. Additionally, metyrapone and ketoconazole can be used as an oral combination therapy to reduce cortisol synthesis synergistically [11]. Etomidate can be used for acute control of severe hypercortisolism by decreasing adrenal steroid production [12].

Surgical excision of the ectopic ACTH-secreting NET enables preservation of adrenal function in the long term despite initial post-operative corticotropic insufficiency. The important pre-operative considerations required include unambiguous identification and localisation of the NET and the absence of distant metastases [1]. Furthermore, identification of patient-specific factors including comorbidities, the effects of cortisol-induced complications, and the fitness for general anaesthesia and surgery must also be considered pre-operatively. After surgical excision of the NET, hydrocortisone replacement, biochemical and clinical monitoring and follow-up with both surgical and endocrinology teams are required to detect recurrence [1].

As shown in Table 1, there are similarities and differences in the diagnosis and management of ectopic Cushing's syndrome of the small bowel and/or mesentery between the various studies. Our method of localisation of the primary NET and subsequent management closely aligns with Singer et al. [13]. However, the patient discussed in Singer et al. [13] had a pituitary lesion excised and a metastatic mesenteric mass resected prior to the discovery of the ileal lesion. The ileal NET was detected by A Ga-69 DOTATATE-PET scan and AN explorative laparotomy. This study again highlighted the difficulty in detecting primary ectopic ACTH-secreting NETs, particularly those of the small bowel.

**Table 1.** Table comparing the localisation techniques and management from case reports of ectopic ACTH-secreting NETs of the small bowel and/or mesentery.

| Location of Ectopic ACTH-Secreting NET | Journal Article | Localisation of Primary NET | Management |
|---|---|---|---|
| Mesentery | Fausshauer et al. [12] | Laboratory: 24 h free cortisol, serum cortisol, plasma ACTH, and the dexamethasone suppression test. Imaging: octreotide scan, CT abdomen/pelvis, 18 FDG-PET, and intraoperative use of a gamma probe radiolabeled 111 In-pentetreotide | Surgical excision |
| Ileal mesentery and liver | Mashoori et al. [14] | Laboratory: 24 h free cortisol, serum cortisol, plasma ACTH, and the dexamethasone suppression test. Imaging: octreotide scan, CT chest/abdomen/pelvis Procedure: liver biopsy, small bowel series, and colonoscopy | Surgical resection of the mesenteric mass and the adjacent small bowel and bilateral adrenalectomy |
| Meckel Diverticulum | Paun et al. [15] | Laboratory: 24 h free cortisol, serum cortisol, plasma ACTH, the dexamethasone suppression test, serum chromogranin A, and urinary 5-hydroxyindoleacetic acid level Imaging: pituitary MRI, CT chest/abdomen/pelvis, octreotide scan, and osteodensitometry | Pre-operative ketoconazole and Sandostin. Surgical resection of the Meckel diverticulum |

**Table 1.** *Cont.*

| Location of Ectopic ACTH-Secreting NET | Journal Article | Localisation of Primary NET | Management |
|---|---|---|---|
| Duodenum and liver | Khare et al. [16] | Laboratory: 24 h free cortisol, serum cortisol, plasma ACTH, and the dexamethasone suppression test. Imaging: pituitary MRI and CT abdomen/pelvis | Surgical resection of the liver and the first part of the duodenum |
| Ileum | Singer et al. [13] | Laboratory: 24 h free cortisol, serum cortisol, plasma ACTH, and the dexamethasone suppression test Imaging: CT chest/abdomen/pelvis, DOTATATE-PET scan octreotide scan, 18 FDG-PET scan, and colonoscopy | Surgical resection of the ileum |

## 4. Conclusions

In summary, this case highlights a diagnostically challenging case whereby imaging and endoscopy/colonoscopy demonstrated a solitary mesenteric mass but intraoperative exploration and histopathology revealed two small bowel NETs as primary lesions. Primary mesenteric NETs are extremely rare with only a few cases reported globally; thus, robust investigations and surgical exploration as both a diagnostic and therapeutic procedure must be considered when searching for lesions.

**Author Contributions:** K.R.C. contributed to the conceptualisation, methodology, resource and data collection, writing the original draft, and the final submission. A.H. contributed to the conceptualisation, methodology, collection of clinical photographs, and the reviewing editing, and writing of the study. A.S. contributed to the conceptualisation, methodology, analysis, data curation, reviewing and editing of the written study, and supervision for the project. All authors have read and agreed to the published version of the manuscript.

**Funding:** This research received no external funding.

**Institutional Review Board Statement:** Ethical review and approval was waived for this study as it is a clinical case report. Only a consent form was required, and this was obtained from the patient.

**Informed Consent Statement:** Informed consent was obtained from all subjects involved in the study.

**Data Availability Statement:** Not available.

**Conflicts of Interest:** The authors declare no conflict of interest.

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
