# Peer review of "A Case of Cushing’s Syndrome from Well-Differentiated Neuroendocrine Tumors of the Small Bowel and Its Mesentery"

_curroncol, doi:10.3390/curroncol30040312_

Round 1

Reviewer 1 Report

In this manuscript, the Authors described the case report of a 68-year-old female with ectopic Cushing’s syndrome due to excessive ACTH secretion from small bowel primary lesions and a mesenteric metastasis.

The case report is well-written and well-structured. However, some parts must be reinforced.

Here I report my suggestions and comments:

-Introduction: I suggest reinforcing this paragraph with more data. Providing more information in the Introduction paragraph gives the reader the right knowledge to interpret the findings reported in the "Case Description". In this direction, I suggest the Authors follow the CARE guidelines (https://www.care-statement.org/).

-Discussion: I suggest also reinforcing this paragraph by providing further perspectives on the value of this case report. In addition, I suggest the Authors create a Table with the previous cases available in the literature with the same diagnosis (Lines 8-9 and Line 149). It would be interesting to compare the findings of the other case report with those reported by the Authors and report these differences in the Discussion paragraph.

-I suggest adding some figures with the microscopic evaluation of the neoplastic tissue removed (if available).

Author Response

Thank you kindly for your valuable feedback
1) In the introduction, I have included more information about the case study topic so the author can interpret findings discussed in the case presentation. I have composed written text in alignment with CARE guidelines. 
2) In the discussion I have provided more information on the value of this case report, focusing on the diagnosis options and challenges for ectopic ACTH producing NETs. I have included a table with previous case reports and their findings regarding investigations and treatment options. 
3) Unfortunately there are no images or figures of microscopic / histopathology specimens available. 
I have also rectified minor spelling errors. 

Reviewer 2 Report

Thank you for inviting me to review the manuscript “A case of Cushing’s syndrome from well-differentiated neuroendocrine tumors of the small bowel and its mesentery".

This topic is interesting, well written and I would like to address several aspects:

1) in the section of case presentation you mentioned that laparoscopic small bowel resection was performed, but in discussion section is presented laparotomy as method of surgical exploration and detection of small bowel lesions.

2) the follow-up period should include the numbers of months for follow-up.

Author Response

Thank you for your review and valuable feedback. 

1) I have ammended this error and deleted the 'laparotomy' statement in the discussion statement. A laparoscopic small bowel resection was performed. 

2) The follow up period was 3 months, I have included this in the case-presentation section. 

Round 2

Reviewer 1 Report

The Authors have addressed adequately my concerns.

Author Response

Thank you for your kind feedback that I had adequately addressed your concerns.